# Towards a UK Airborne Bioaerosol Climatology: Real-Time Monitoring Strategies for High Time Resolution Bioaerosol Classification and Quantification

Ian Crawford [1,*], Keith Bower [1], David Topping [1], Simone Di Piazza [2], Dario Massabò [3], Virginia Vernocchi [4] and Martin Gallagher [1]

1 Department of Earth and Environmental Sciences, The University of Manchester, Manchester M13 9PL, UK
2 Laboratory of Mycology, Department of Earth, Environment and Life Sciences, University of Genova, Corso Europa 26, 16132 Genoa, Italy
3 Department of Physics, University of Genova, INFN-Division of Genova, 16146 Genoa, Italy
4 National Institute of Nuclear Physics (INFN)-Division of Genova, 16146 Genoa, Italy
* Correspondence: i.crawford@manchester.ac.uk

**Abstract:** Biological particulate matter (BioPM) is a poorly constrained, ubiquitous, and diverse subset of atmospheric aerosols. They influence climate, air quality, and health via many mechanisms, spurring renewed interest in constraining their emissions to elucidate their impacts. In order to build the framework required to assess the role of BioPM in these multidisciplinary areas, it is necessary to develop robust, high time-resolution detection methodologies so that BioPM emissions can be understood and characterized. In this study, we present ambient results from intensive monitoring at UK peri-urban and coastal ground sites using high time-resolution real-time bioaerosol spectrometers. We demonstrate the utility of a new dimensional reduction-driven BioPM classification scheme, where laboratory sample training data collected at the ChAMBRe facility were used to generate broad taxonomic class time series data of key species of interest. We show the general trends of these representative classes, spanning spring, early summer, and autumn periods between 2019 and 2021. Diurnal behaviors and meteorological relationships were investigated and contextualized; a key result arising from this study was the demonstration of rainfall-induced enhancement of nighttime *Penicillium*-like aerosol, where rainfall crucially only acts to enhance the quantity emitted without significantly influencing the early morning timing of peak spore liberation.

**Keywords:** bioaerosols; air quality; real-time detection; UV-LIF spectroscopy; machine learning

## 1. Introduction

Biological particulate matter (BioPM) accounts for over 10% of the continental supermicron aerosol number concentration and more than 25% of global organic aerosol emissions [1,2]. It represents a diverse and complex classification of aerosol, comprised of viruses, bacteria, fungi, and pollen; these aerosols span sizes ranging from 10s nanometers to up to 100 μm, and they can be transported large distances, remaining airborne for long periods of time [3–6]. BioPM is of increasing interest to the atmospheric science community due to its potential impact on climate via cloud–aerosol interactions and air quality and for understanding pathogen and allergen dispersal.

BioPM has many deleterious impacts on public health, ranging in severity from allergies to death, causing personal and economic harm. Fungal spores pose a significant and often underappreciated persistent threat. It is estimated that 1.5 M annual deaths are caused by invasive fungal diseases, impacting the immunocompromised, cancer patients, the elderly, and those undergoing intensive care treatments [7]. The UK has one of the highest prevalences of diagnosed asthma, affecting ~10% of the population, causing approximately 1400 asthma deaths annually [8]. Due to difficulties in amassing appropriate datasets for

collation, the fungal contribution towards the UK's death rate has not been established; however, a relationship between elevated summertime fungal emissions and adult mortality has previously been observed [9,10]. There is also increasing speculation that the allergenicity of BioPM may be enhanced via interactions with urban pollution, furthering the need to quantify emissions if urban exposure impacts are to be elucidated [11]. Many plant and animal diseases are spread by fungal spores, resulting in significant economic damage. Fisher et al. [12] estimated that 64% of severe crop die-offs and extinctions are caused by fungal pathogens, posing a persistent and growing threat to food security; they also demonstrate indirect harm pathways, with the US experiencing agricultural losses of USD 3.7 B per year though increased crop pests as a consequence of a rapidly declining bat population due to invasive fungal diseases (e.g., white-nose syndrome [13]).

Clouds play an important role in regulating the Earth's energy balance via their interactions with solar and terrestrial radiation. Primary ice formation at temperatures warmer than $-38\,°C$ requires the presence of a catalyst ice nuclei (IN) particle via heterogeneous processes [14,15], where certain species of BioPM have been identified as potent IN in the critical $-15$ to $0\,°C$ warm regime owing to the presence of ice nucleating proteins. Fungi such as *Fusarium* have displayed ice activity as warm as $-3.5\,°C$ [16], and such fungi have been observed in the upper atmosphere and free tropospheric cloud water samples [17–19]. Significantly, only small quantities ($\sim 0.01\,\text{L}^{-1}$) of such warm temperature IN are needed to induce rapid mixed-phase cloud glaciation via secondary ice processes, resulting in reduced cloud lifetime and reflectivity, ultimately impacting the radiation budget [20]. More broadly, the *so-called* bio-precipitation hypothesis postulates that the emission of ice-active BioPM is enhanced by rainfall (e.g., as demonstrated by Huffman et al. [21] and Niu et al. [22]), resulting in an enhancement of rainfall via ice processes; it is suggested that this, in turn, fosters an environment beneficial to the growth of plants and microorganisms, thus yielding further enhancement of ice-active BioPM [23,24]. Understanding emissions of ice-active BioPM is critical to elucidating their climate impacts and also how BioPM emissions may be impacted by a changing climate and future land use strategies.

### 1.1. Detection Methods

The detection and classification of BioPM remain a notable technical challenge, where tradeoffs between taxonomic specificity and time resolution are common experimental burdens [25]. There is a wide range of traditional active and passive air sampling systems, which collect ambient aerosol onto a substrate for subsequent offline analysis, where each system is selected to meet experimental design goals [26]. Once collected, the samples can then be analyzed via many methods (e.g., qPCR, microscopy, and metabarcoding) depending on the level of quantification and specificity required [26]. Sample integrations are typically long (>24 h), which makes investigating dynamic emissions processes difficult. International standards are generally not prescriptive, making comparison of results between studies challenging. Furthermore, different systems may display significant biases and sampling inconsistencies [27]; as such, there is no widely accepted way to interpret data outputs for exposure estimation [28].

Over approximately the last 15 years, ultraviolet light-induced fluorescence (UV-LIF) spectrometers have been developed to detect and classify BioPM in real time. Their working principle exploits the intrinsic autofluorescence of biological particles to segregate and classify them from background aerosols; many systems work on a single particle basis, with integration periods of 1–5 min being typically possible depending on ambient aerosol concentrations. Older discrete detector systems, such as the wideband bioaerosol spectrometer (WIBS) and UV-APS, lack the spectral resolution and morphological interrogative capability to unambiguously classify biological aerosols [25,29]; however, more sophisticated spectrometers boasting enhanced spectral resolution and particle shape information are becoming available, increasing confidence in discriminative capability [30–33]. While it is expected that real-time classification capability will continue to improve with spectrometer and classification scheme development, these methods will likely lack the deep specificity

afforded by offline methods for some time. The value of real-time systems lies in their inherent high-time resolution capability and low deployment overheads, making them ideally suited for long-term monitoring and for investigating rapid and dynamic changes underpinning dispersion and exposure.

### 1.2. Aims and Objectives

The BIOARC project (Towards a UK Airborne Bioaerosol Climatology) aimed to evaluate the airborne concentrations of BioPM in the UK via targeted aircraft and ground-based sampling; early emerging results from the aircraft campaign were presented by Song et al. [18] who demonstrated that *Cladosporium* was a major component in the boundary layer via PCR analysis. In this study, we present an overview and synthesis of the UK-based ground site UV-LIF bioaerosol spectrometer datasets collected during BIOARC. We demonstrate the utility of a new machine learning classification scheme; we then use classifier outputs to generate broad class time-series data products to investigate diurnal behavior and meteorological relationships. In summary, the key aims are:

1. Demonstrate the performance of a new dimensional reduction-based real-time BioPM classifier.
2. Deploy the real-time system to characterize and quantify seasonal BioPM at two UK sites of interest.
3. Investigate the influence of environmental factors on BioPM emission.

A key result emerging from this study is evidence of rainfall-induced enhancement in the nighttime *Penicillium*-like aerosol concentration, where crucially, while concentrations are enhanced compared to dry periods, the timing of the emission remains stable around the early hours of the morning.

## 2. Methods

### 2.1. The Multiparameter Bioaerosol Spectrometer

The multiparameter bioaerosol spectrometer (MBS) is a biofluorescence spectrometer developed by the University of Hertfordshire to detect and classify bioaerosols in real-time time via the collection of autofluorescence spectra, size, and morphological parameters on a single particle basis. Full technical descriptions of the MBS were provided by [29,30,34] and a brief description is presented below. Similar to its predecessor, the wideband integrated bioaerosol sensor, the MBS' principle of operation revolves around the deep UV excitation of a single aerosol particle within the sensing region with an optically filtered xenon flashlamp (280 nm), where any resultant autofluorescence is then detected over 8 bands spanning a range of 315–640 nm via a grating spectrometer and multichannel photodetector. This configuration was chosen to capture the autofluorescent emission bands of several key bioflurophores commonly found in bacteria, fungi, and pollen. The targeted fluorophores and their approximate emission maxima and ranges at an excitation of 280 nm are as follows [32]:

- Tyrosine: $310 \pm 20$ nm;
- Tryptophan: $365 \pm 40$ nm;
- Riboflavin: $520 \pm 30$ nm;
- Chlorophyll b: $640 \pm 10$ nm.

Note that 315–640 nm denotes the mid-channel values; thus, the actual span of the detection range of the instrument is slightly wider. Each channel spans approximately 50 nm; thus, the upper limit is 665 nm and captures chlorophyll b. The lower limit is set at 310 nm via a long-pass filter preceding the diffraction grating, but crucially still captures tyrosine emission. This enhanced spectral resolution enhances discriminative capability over the WIBS spectrometer [29,30].

Aerosols are drawn into the sample volume at a sample flow of 0.2 L per minute (LPM), where they are constrained within a HEPA-filtered sheath flow (1 LPM, resultant total inlet flow 1.2 LPM) to minimize optical contamination and provide a well-collimated aerosol

beam for the detection system. Aerosols are first detected and sized over a range of 0.5 to 15 μm in diameter with a 12 mW 635 nm laser; upon detecting a particle within this size range, a 250 mW 637 nm laser is triggered to illuminate the particle with sufficient intensity to interrogate particle morphology via a dual complementary metal-oxide semiconductor (CMOS) array image sensor. This dual 512-pixel array collects scattered light from the illuminated particle, providing two sectional chords through the 2D spatial scattering pattern. Several characteristic morphological parameters are then generated from CMOS distributions at acquisition which are described in detail by Crawford et al. [29]. Briefly, these parameters provide various proxies for particle symmetry and aspect ratio. 10 μs after particle detection, the xenon flashlamp is triggered, and any resultant autofluorescence is focused onto the detection optics with two hemispherical mirrors. The xenon flashlamps are capable of a maximum strobe rate of ~125 Hz, limiting the maximum acquisition rate. In practice, the instrument rarely approaches this limit during ambient sampling due to the instrument's relatively coarse detection range.

Fluorescence is determined by periodically running the instrument in so-called forced trigger (FT) mode, where the flashlamps are strobed at 10 Hz in the absence of particles to determine the baseline fluorescence of the detection volume. The mean value of the FT spectral data + 7 standard deviations (typically referred to as 7σ in UV-LIF spectrometer literature) is used to define a threshold value for each channel, which is subsequently subtracted from the acquisition data, where negative values are clipped at zero. The high value of 7σ is chosen as it typically retains true, highly intense biofluorescence from biological particles while rejecting typically weakly fluorescing non-biological interferent particles, thus naturally rejecting interferent particles from the fluorescent population [35]. For a particle to then be considered fluorescent, it must display non-zero values in 2 or more channels to ensure that the signal has not arisen from spurious reflections within the system [32].

### 2.2. Site Descriptions

UV-LIF spectrometers were deployed at several strategic UK ground sampling sites, which are now described.

### 2.2.1. Cardington

The MBS was deployed at the UK Met Offices' Meteorological Research Unit, Cardington, Bedfordshire, UK (52.104° N, 0.421° W, 29 m ASL). Briefly, the site is host to several key standard meteorological measurements and extended surface and balloon-borne observations. The MBS sampled ambient air via a total inlet at an approximate height of 3 m above ground level for 71 days between 11 April 2019 and 19 June 2019, capturing the end of the spring fungal emission season [9]. The Cardington site is rural to peri-urban in nature and is surrounded by farmland. The towns of Bedford and Kempston lie to the northwest and west, respectively, approximately 5 km away.

### 2.2.2. Weybourne Atmospheric Observatory

An MBS was deployed at key periods in 2020 and 2021 at the Weybourne Atmospheric Observatory (WAO). WAO is a Global Atmospheric Watch regional station located on the North Norfolk Coast, UK (52.951° N, 1.122° E, 15 m ASL). The site collects routine meteorological data alongside high-precision trace-gas and chemistry measurements. The bioaerosol spectrometers were located on the roof terrace of the observatory in custom weatherproof enclosures, sampling at an approximate height of 5 m above ground level. The WAO site is rural coastal and situated ~0.2 km from the coast, which runs approximately WNW to EbS at this location. Inland lies farmland and wooded areas. The nearest significant towns are Sheringham (6 km to the East) and Holt (5 km SSW).

The WAO site deployments were as follows:
1. MBS, 15 September 2020 to 3 November 2020 (50 days);
2. MBS, 15 April 2021 to 16 July 2020 (93 days).

### 2.3. Classification Method

In this manuscript, we use uniform manifold approximation and projection for dimension reduction (UMAP) [36] to classify the MBS data into broad representative BioPM classes. UMAP is a dimensional reduction technique which is generally used to visualize high dimensional data or as a pre-processor for other machine learning techniques, but it also performs excellently as a robust classifier, as will be demonstrated. Here, it is used to compress the 14-dimensional spectral and morphological MBS data into a 2D representative space, where the UMAP transformer is optimized with labeled training data, allowing the model to generate a transformed space with high spatial separation between the classes; this model can then transform unseen ambient data into the same space to aid classification. The principal methods used to train the model are:

1. Characterizing the autofluorescent emission of each test species over 8 narrow bands between 315 and 640 nm after deep UV excitation at 280 nm. This probes the relative biofluorophore makeup of the bioaerosol under test.
2. Characterizing particle morphology via interrogating two chords of the 2D scattering image with a dual CMOS detector. This delivers proxy information on morphological features such as particle sphericity/aspect ratio and surface roughness. Additionally, particle size is determined via Mie scattering.

This provides the UMAP model with 14 parameters in total to characterize the bio-autofluorescence, size, and shape of each training species.

As described by Crawford et al. [29], prior to training a classifier, it is necessary to pre-process the raw instrument data in order to increase the selectivity of the classifier [37]. Only particles above a $7\sigma$ threshold are retained for training, and a particle must exhibit fluorescence in at least 2 channels to be considered fluorescent to filter out spurious measurements and noise [32]. As UMAP performs best when each input parameter is on approximately the same range and scale, all parameters are normalized by their maximum expected value, e.g., the fluorescence detectors saturate at approximately a value of 2200 arbitrary units, and the maximum particle size detection limit is 15 μm; thus, these values are used to scale the parameters between 0 and 1. A full description of the pre-processing method is provided at the end of Section 2.1.

The UMAP transformer model was trained using laboratory-generated bacteria, fungal spores, and pollen samples. The bacterial and fungal samples were collected at the ChAMBRe simulation chamber facility (Istituto Nazionale di Fisica Nucleare, Sezione di Genova) as part of the EU-H2020 ATMO-ACCESS Trans-National Access programme. The ChAMBRe is a large stainless-steel chamber (approx. 2.2 m$^3$) custom-built to perform studies on bioaerosols. The chamber is cleaned in-between samples by venting to a pressure of $10^{-2}$ mbar and replenishing with filtered ambient air. The chamber was also routinely sterilized via the generation of ozone with a high-powered UV lamp. Bacterial samples were aerosolized with a SLAG nebulizer (CH-Technologies) in Milli-Q water [38]; fungal samples were dry dispersed into the chamber with dry, filtered compressed air. A full technical overview of the facility was provided by Massabò et al. [39]. Gram-positive *B. subtilis* and Gram-negative *E. coli* were selected to represent ambient bacteria; *Penicillium* and *Cladosporium* were chosen to represent ambient fungal spores. *Alternaria* was also investigated; however, owing to its large size (typically >15 μm in length), this sample was difficult to adequately aerosolize into the chamber for sampling. Additionally, nettle pollen samples from the Dstl aerosol challenge simulator facility were also included in the training data set (see [29,34] for experimental details). Several other pollens were tested, but they typically exceed the instrument's upper size limit of ~15 μm and thus are not included in the training dataset.

## 3. Results and Discussion

### 3.1. UMAP Classifier Training and Performance

A statistical overview of the MBS CMOS shape parameters, size, and autofluorescent spectra for each sample is provided in Figure 1. Each broad taxonomic class displays

unique characteristic features, e.g., the combined bacterial sample class is notably smaller, and its spectral mode occurs in a lower channel than that of the fungal spore classes; nettle pollen tends to be highly fluorescent and morphologically distinctly different to the other classes.

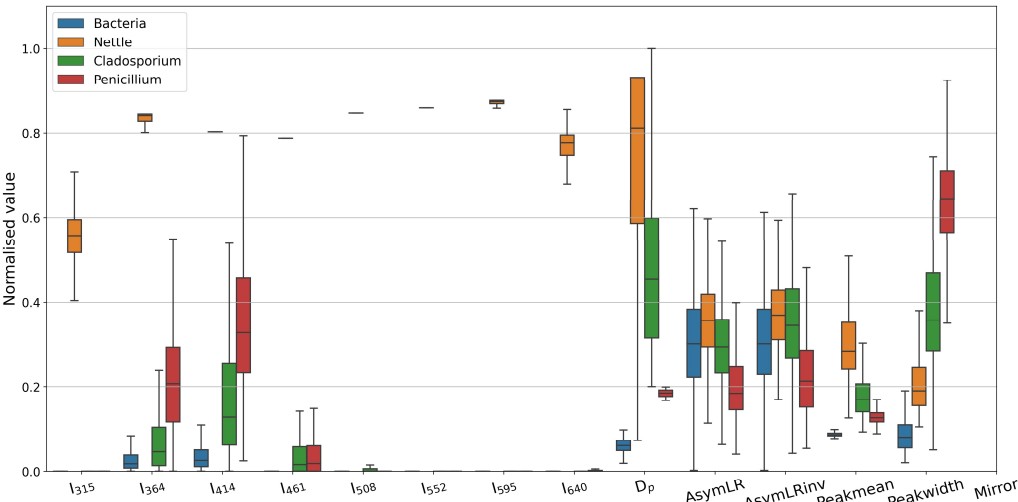

**Figure 1.** Training data overview. Box and Whisker plots of each broad class are shown; whiskers denote the 5th and 95th percentiles. Data are normalized by the maximum expected value for each parameter. $I_x$: fluorescent intensity at wavelength x [nm]; Dp—particle diameter; AsymLR—symmetry between left and right CMOS arrays; AsymLR: as AsymLR, but with the right array inverted; Peakmean: ratio of peak to mean CMOS array values; Peakwidth: estimate of the mean width of the array, defined as mid-point between mean and peak values; Mirror: measure of symmetry between top and bottom half of each array.

The greatest similarity exists between the *Penicillium*-like and *Cladosporium*-like spectra, which is to be expected as they are both fungal species. This may result in some conflation between the two; thus, caution may be required when using these classification products in finer granularity beyond a general fungal classification. In the outdoor ambient environment, one would expect to see different timing trends in the diurnal maxima of each of these classes, relating to their evolutionary optimised dispersal mechanisms and airborne survivability to maximize colonization [40,41]. Some similarities in the second fluorescent channel and CMOS morphological parameters are seen in the bacterial and *Cladosporium* training data; this may result in some misclassification. This likely represents the limit of specificity that the MBS is capable of as the relatively coarse spectral resolution may contain emissions from several biofluorophores in a single channel; it is envisaged that increased spectral resolution and/or additional excitation bands would enhance capability. We caveat that the classifier outputs may be conflated with other, potentially unknown, species by referring to the output products as being species-like (e.g., *Penicillium*-like) to highlight that the classified particles are similar in characteristics to their species training data counterparts but may not necessarily be the same species. Similarly, conflation between fungal species such as *Penicillium* and *Aspergillus* is commonplace with other sampling methods, e.g., spore trap microscopy [9].

To assess the performance of the model, the training data are split into a training subset (20%) and a testing subset (80%) and then the transformer is applied to the testing subset to examine if the data points are correctly placed within the 2D space. Two-dimensional UMAP dimensional reduction using 10 nearest neighbors was applied to the training subset and is shown in the left panel of Figure 2. Generally, good separation between the classes can be observed in the transformed space. We define an approximate classification boundary for each class as two times the mean of the standard deviations of the x and y components of each class, centered on the mean value. This is shown as a circle encompassing each

classification and is calculated for the training subset data only. The right-hand panel of Figure 3 shows the results of transforming the remaining test subset data into the trained 2D space, where generally, it can be seen that the majority of the test data points reside within their respective $2\sigma$ classification boundaries while also demonstrating the potential conflation pathways between some fungal species.

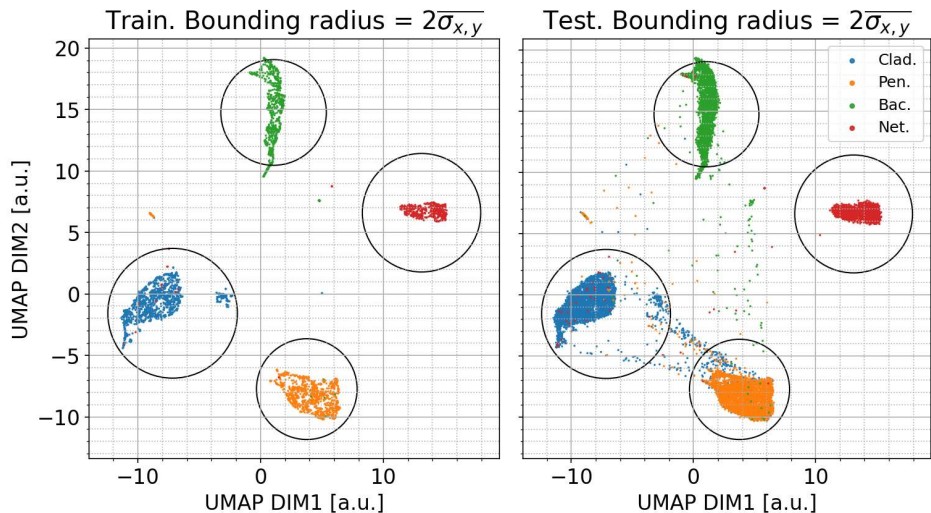

**Figure 2.** UMAP dimensional reduction of the training data. Left panel: transformer model trained on 20% of the training dataset; right panel: trained transformer model applied to the remaining test subset of the training dataset. A bounding radius of 2 times the mean of the standard deviations of the x and y components of the training set data for each class is shown as a guide perimeter to define class boundaries.

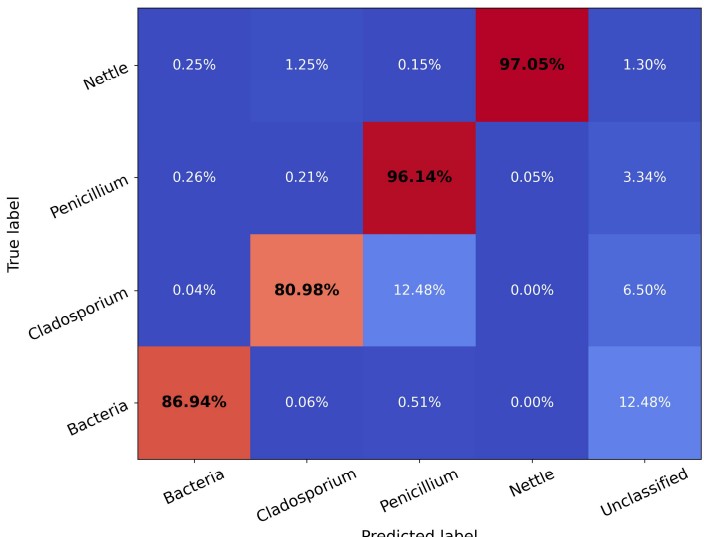

**Figure 3.** Confusion matrix of the UMAP classification model using the testing set grouped into broad classes using a $2\sigma$ boundary (see text). The proportion of the model-predicted labels (columns) are compared to the true label (rows) for each broad training class and presented as a percentage value.

Figure 3 shows a confusion matrix assessing the UMAP transformer performance. Generally, it can be seen that the model performs exceptionally well, classifying the reserved testing subset of the data to a high level of accuracy. The conflation between the fungal classes was not unexpected and is not problematic as they represent the same overarching taxonomic kingdom. This provides us with confidence that the fungal classifier is

likely to represent ambient fungal material; however, the bacterial classifier may contain misclassified fungal material, so caution must be applied when interpreting results. It may be necessary to perform further analysis on the bacterial classifier results to resolve the two classes. It is likely that other supervised learning methods would also suffer from such conflation as they would similarly be limited by the spectral resolution of the MBS, e.g., Crawford et al. [29] demonstrated the conflation between bacterial and fungal classes using a gradient-boosting classifier on MBS data. While at first glance, the classification accuracy may seem less impressive than other methods such as gradient boosting (e.g., Ruske et al. [30]), the high scores attained by these methods may result from overtraining, making the resultant model prone to misclassification.

One advantage of the UMAP method employed here over many other supervised methods is the freedom to define classification boundaries, allowing for flexibility when dealing with the limitations of the scope of the training data being too narrow (e.g., over-training due to the class data not being general enough) or shifts in parameters due to instrument drift over time. It may even be possible to apply a trained model to different MBS instruments, provided their responses are broadly similar. A further advantage of this method is that it naturally highlights data which fall outside of the training classifications, allowing for the ready creation of an unclassified class; this is not always easy to generate with other methods such as gradient boosting.

We now apply the transformer to the Cardington ambient data to examine if the classifier can produce results comparable to the offline data, as failure to do so would suggest that the model is inadequate. Figure 4 shows a density scatter plot of the transformed ambient Cardington data, where it can be seen that hotspot clusters fall within the bounding radius for each classification as expected.

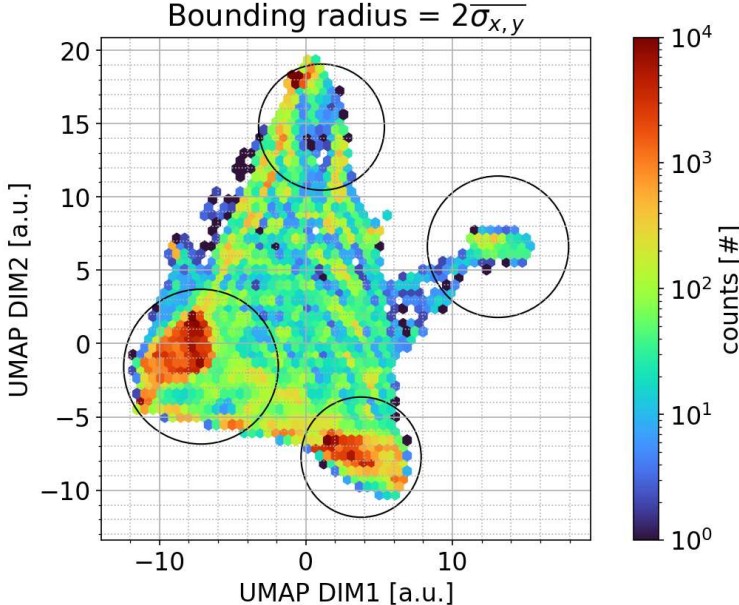

**Figure 4.** Two-dimensional density scatter plot of the ambient Cardington data in the transformed space. The 2σ bounding radius from Figure 2 is shown to demark the classification boundaries.

The demarked ambient data for each class at each sampling site are compared to the training data in Figure 5, where generally, good agreement between the ambient and training data can be observed for the fungal and nettle classes. Crucially, there is consistency between the sites, providing confidence that the instrument is detecting and classifying similar particles at each site. The nettle pollen class is generally highly fluorescent in all channels, such as the training data, and features similar morphological parameters.

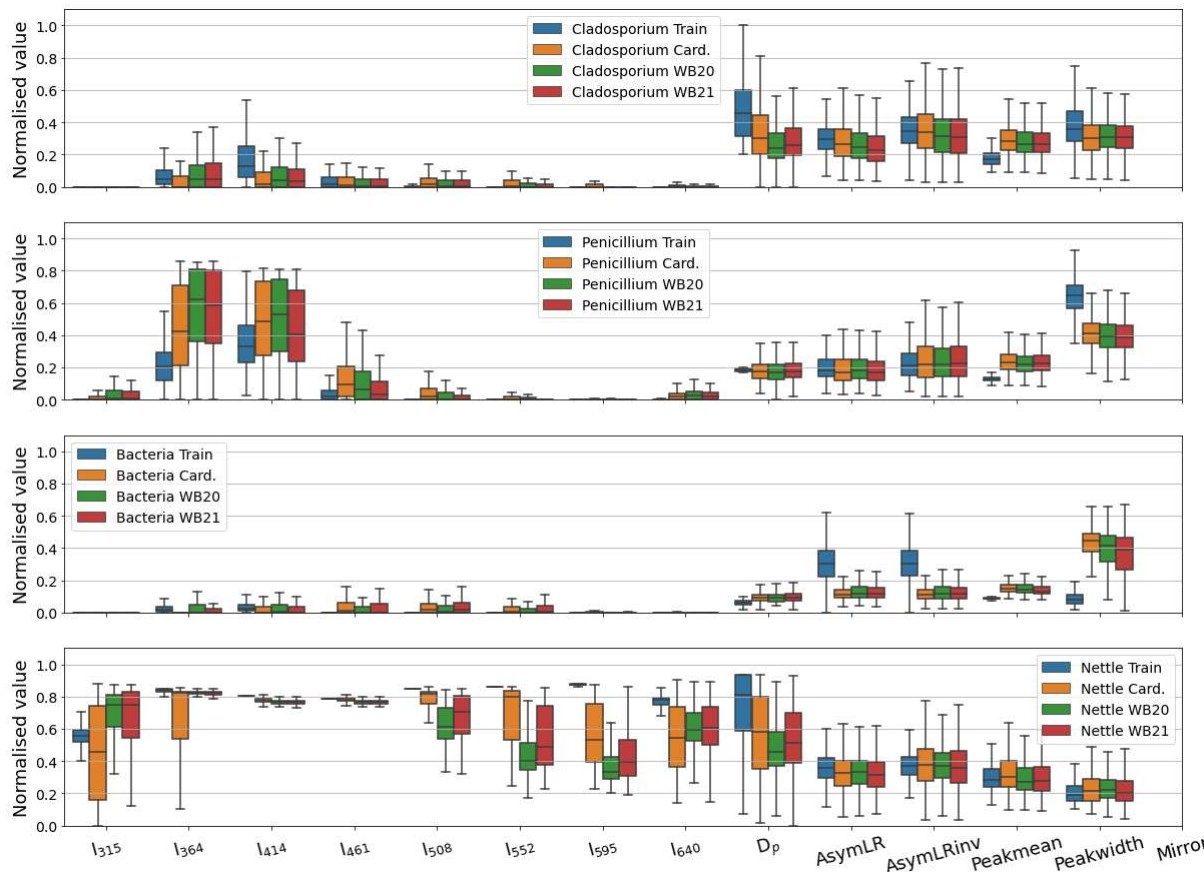

**Figure 5.** Same as Figure 1 but compares the ambient and training parameters for each classification at each sampling site.

The bacterial class is dissimilar to the training data, featuring fluorescence beyond the third channel of the MBS, as well as morphological differences. The extended fluorescent emissions to longer wavelengths are suggestive that the bacterial class also contains misclassified fungal-like particles (or some other unidentified class), and further analytical intervention is required to build an accurate bacterial classifier. From careful inspection of the bacterial training data, it is clear that the training aerosol rarely displays fluorescence in the fifth to eighth channels; we exploit this to remove likely non-bacterial particles from this class by excluding any particles initially classified as bacteria which display any fluorescence in these upper channels. When inspecting the remaining bacterial subset after applying this filter, some particles exhibit fluorescence in the fourth channel; this may be due to a slight drift in the optical alignment, the emission characteristic of different species, or as a result of atmospheric or other processing. However, the modal fluorescence should not occur in this channel, so we also include an additional filtering step where the intensity of the fourth channel must be less than that of the third channel. Particles which fail either criterion are retained as a separate class for further inspection but are merged with the general unclassified BioPM class in the analysis presented in this manuscript.

### 3.2. Cardington 2019

The MBS was deployed at Cardington during late spring and early summer of 2019, well placed to capture the end of the typical spring fungal emission maxima as it transitions into summer [9].

Figure 6 shows a time series of the standard and UMAP classifier data products for the deployment period, alongside accompanying meteorological measurements. Generally, the

fluorescent number concentration is relatively stable throughout the measurement period, while the non-fluorescent concentration displays a slight decrease over time.

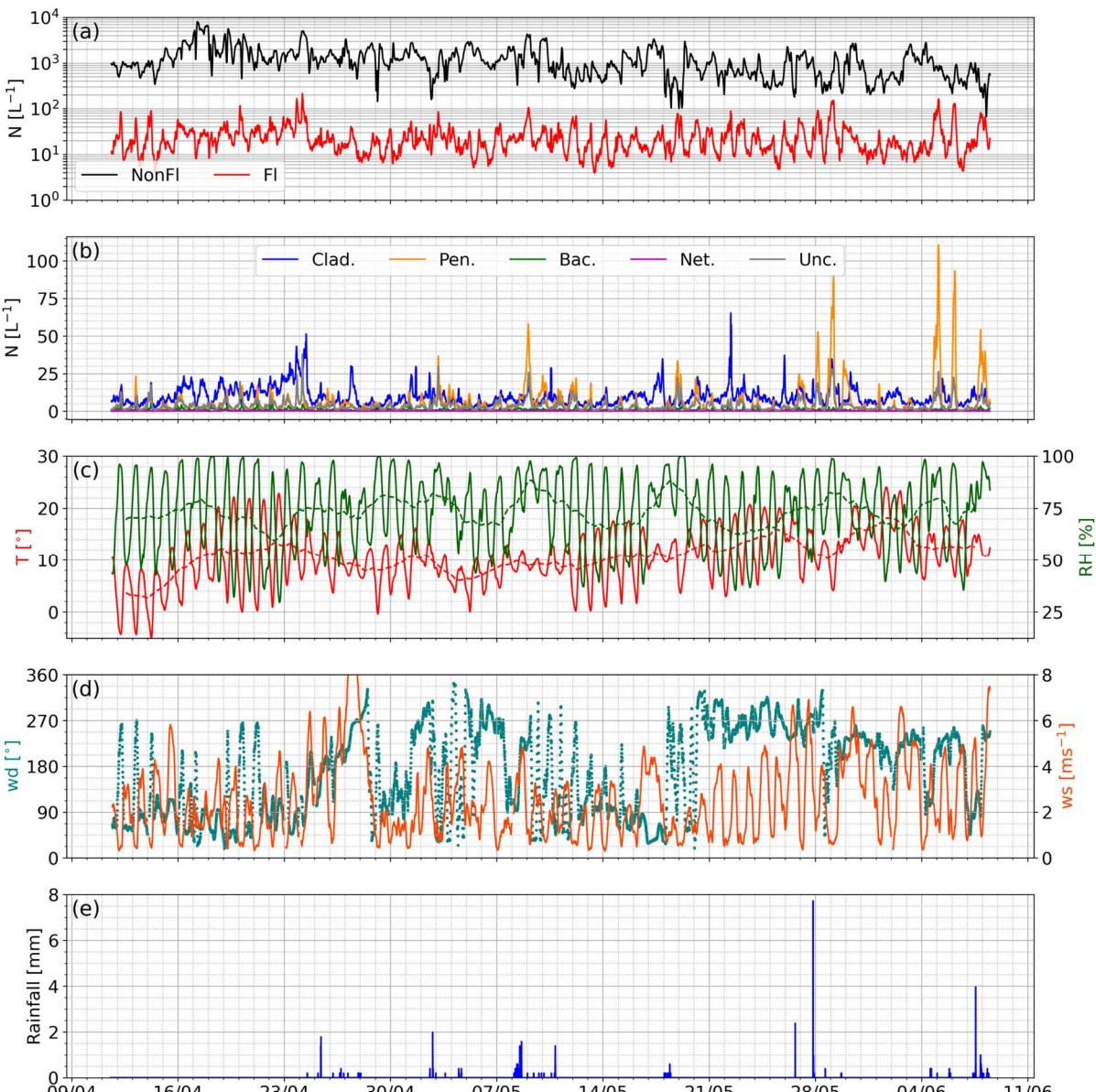

**Figure 6.** Timeseries of MBS data products and meteorological parameters at the Cardington site. (**a**) Non-fluorescent and fluorescent concentrations; (**b**) UMAP classifier product concentrations; (**c**) air temperature (red, left axis) and relative humidity (green, right axis); (**d**) wind direction (blue, left axis) and speed (orange, right axis); (**e**) rainfall. A 3 h rolling mean was applied to all MBS data to improve clarity. A 24 h rolling mean is also shown for temperature and relative humidity to highlight the general trend in these parameters, depicted by a dashed trace.

The filtered bacteria-like class displays low concentrations throughout the measurement period, with a weak diurnal maximum peaking later in the morning than the fungal classes. *Cladosporium*-like particles display broad, sweeping variation, while the *Penicillium*-like and unclassified particles show a similar strong diurnal variation, peaking in the early morning. This diurnal behavior will be investigated and compared to the other sites in a later section. There are also several strong, episodic diurnal events with peak concentrations of up to ~90 L$^{-1}$, which tend to occur after rainfall events. Figure 7a investigates this in greater detail, zooming in on the period 26 May–9 June, where several rainfall-enhanced

diurnal maximums occur. It can be seen that the maximums still tend to occur around their typical early morning hours, ranging from 3 to 15 h after a rainfall event. However, even modest rainfall can promote the enhancement of nighttime *Penicillium*-like emissions; no clear relationship between the rainfall amount and emission strength was observed. We speculate that either the emission enhancement mechanism is independent of the rainfall intensity, e.g., surface wetting effects, or simply that while the rainfall intensity measured at the site is indicative of a rain event in the general area, it does not necessarily represent the intensity at the emission source, thus does not correlate at the point of detection. The quiescent period 31 May–4 June features no rainfall, where no enhanced *Penicillium*-like concentrations were observed, further suggesting that rainfall is a significant driver for the observed diurnal enhancement. We further investigate the impacts of rainfall by segregating wet and dry periods, whereas before, a wet period was defined as the 15 h following the onset of rain. Figure 7b shows box and whisker plots summarizing the concentrations of the different BioPM classes during wet and dry conditions, where it can clearly be seen that *Penicillium*-like particles are significantly enhanced during wet periods. The other classes are largely unaffected, while a small increase in unclassified particles is observed during wet conditions.

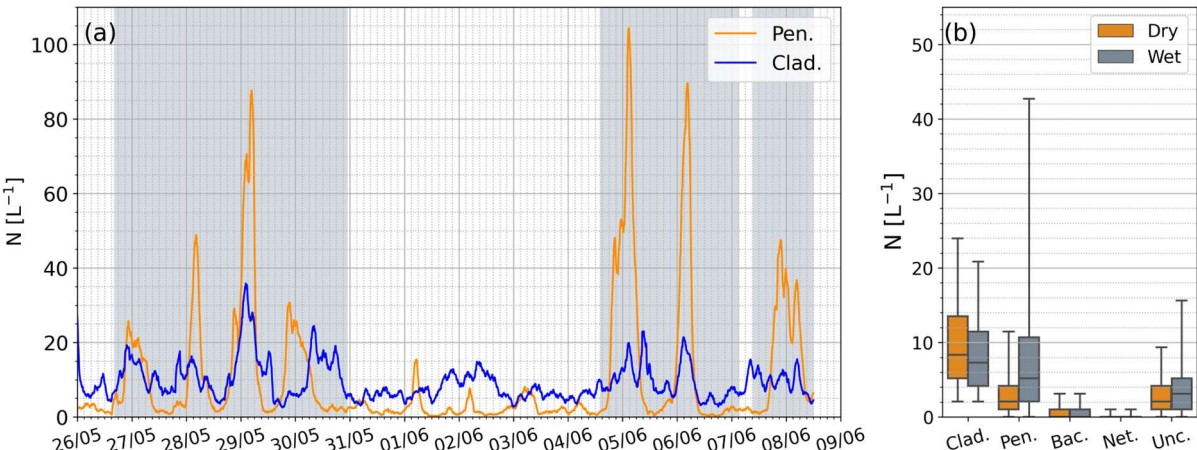

**Figure 7.** Impact of rainfall at Cardington. (**a**) Zoomed timeseries of MBS fungal-like concentrations. Shaded regions indicate a 15 h period from the onset of a rainfall event. Note that multiple rain events may create overlapping shaded regions. (**b**) Box and whisker plots of the different class concentrations under wet and dry conditions. Wet conditions are classified as a 15 h period from the onset of a rain event. Whiskers denote the 5th and 95th percentiles.

### 3.3. Weybourne 2020 and 2021

The MBS was deployed at WAO during the autumn of 2020 and the end of spring and early summer of 2021, capturing the end of the *Cladosporium* season prior to the transition to winter and the October *Penicillium* maximum, and the early spring to summer fungal season transition, respectively [9]. The deployments at WAO were restricted due to local and governmental COVID-19 policy in place at the time, thus did not fully capture the inter-seasonal changes due to the necessity to work around the fluid nature of these restrictions. Figure 8 shows a time series of MBS and meteorological parameters for the 2020 and 2021 deployments, where a low background of bacteria-like and nettle-like particles were observed during each period; the fungal class concentrations were much higher than those of the bacteria- and nettle-like classes, with broader *Cladosporium*-like emissions detected and episodic diurnal *Penicillium*-like emissions displaying characteristic early morning peak concentrations. Unfortunately, no co-located contemporaneous high time-resolution rainfall data were available for the site at the time of deployment; however, hourly data from the nearby Weybourne Met Office monitoring site were available via the Met Office Integrated Data Archive System (MIDAS) [42]. While this dataset is of limited temporal

resolution, it demonstrates that rainfall occurred on most days during the 2020 autumn deployment. Furthermore, it is noteworthy that days displaying minimal *Cladosporium-* and *Penicillium*-like concentrations feature either no or very little rainfall, further suggesting that rainfall plays a significant role in fungal-like aerosol emission.

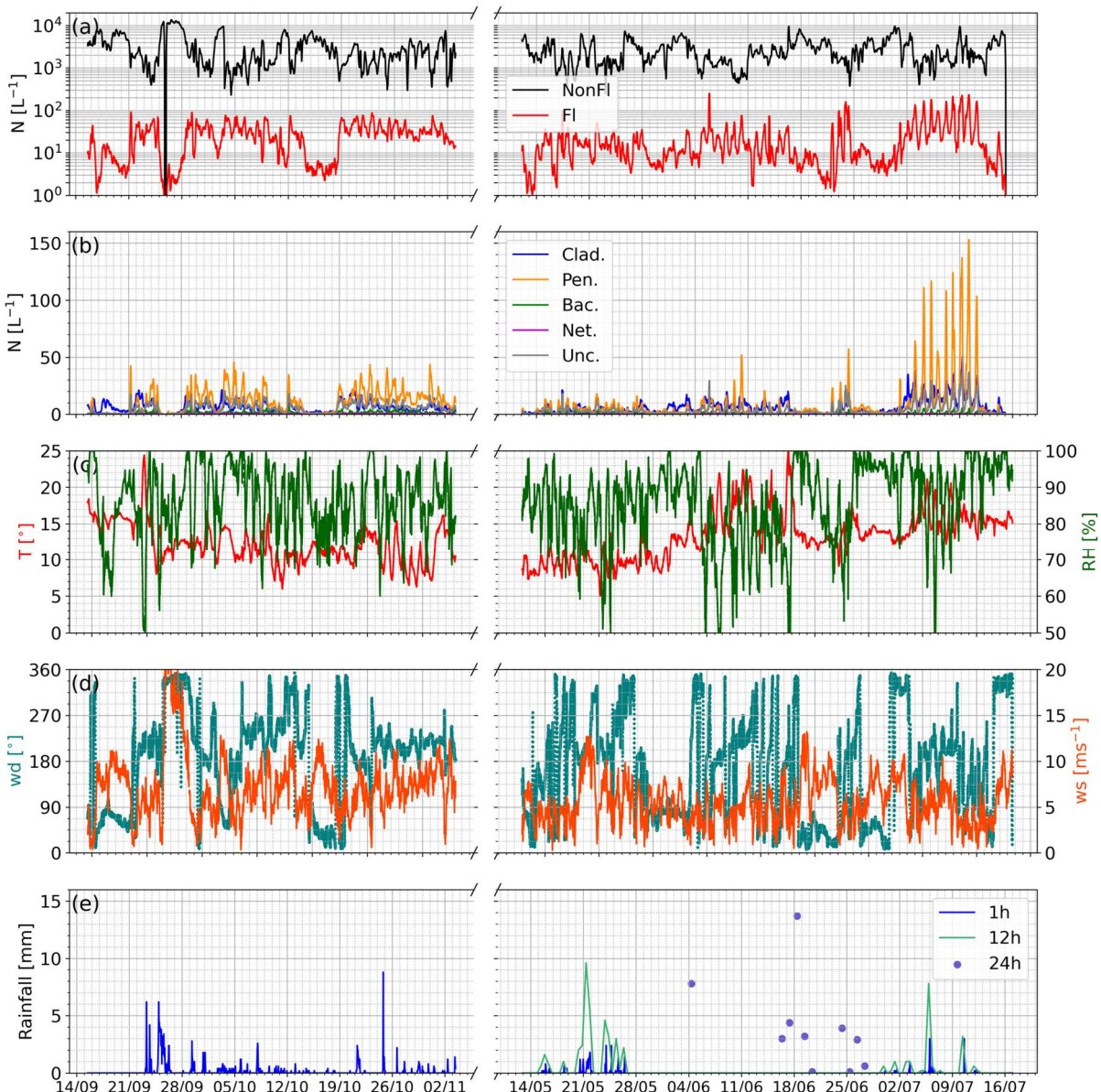

**Figure 8.** Same as Figure 6, but for Weybourne 2020 and 2021 deployments. Note: Rainfall data provided via Met Office Integrated Data Archive System; period 09:00 4 June 2021–10:00 29 June 2021 features only sparse 12 and 24 h rainfall integrations, which, while offering limited coverage, are included for reference. Additionally, note the *x*-axis break, where the 2020 deployment is shown on the left of the break, and 2021 on the right to aid comparison of the two deployments.

During the 2021 spring–summer deployment, the *Penicillium*-like class again displayed diurnal nighttime emissions, and the *Cladosporium*-like class displayed broader general emission events. Towards the end of the deployment, there were 11 consecutive days featuring significantly greater fungal emissions, with diurnal maximums of 50 to 150 $L^{-1}$, a value much greater than observed during previous deployments. Unfortunately, quality-assured hourly rainfall data were not available via MIDAS for the majority of June; thus, it is necessary to utilize 12 and 24 h rainfall integrations, both of which feature significant outages during this period. While the period of fungal enhancement in July coincides with

some rainfall events, it is difficult to ascertain if the rainfall was driving the enhancement as several significantly enhanced nighttime emissions events occur in the absence of rain. The paucity of reliable high time-resolution rainfall data during June to interrogate the enhanced fungal events further makes interpreting the impact of rainfall difficult; thus, the effects of rainfall remain ambiguous during this deployment. Generally, the observed seasonal trends of fungal emissions are consistent with previous studies investigating seasonality. For example, Sadyś et al. [9] compiled 5 years of UK spore trap data to demonstrate that *Penicillium* displays peak emissions during October, with subsequent minimum activity in spring, which is constant with our results. Similarly, they demonstrate a peak in *Cladosporium* activity during July, which is mirrored in our observations. While different in scope to our study, recent work by Maki et al. [43] demonstrated an autumnal maximum in *Penicillium*, again consistent with our real-time observations. The seasonal trend in bacteria is not as clear, with the baseline being similar in all cases; this is consistent with the findings of Maki et al. [43], who demonstrated that there were no apparent seasonal changes in bacterial composition, suggesting that any changes were due to daily variations in environmental conditions.

### 3.4. Site Synthesis and Relationships

We now move to a broader synthesis analysis and comparison of the site datasets to investigate general behaviors and relationships. Figure 9 compares the generalized concentrations of key classes at each site, where the data for each deployment have been overlaid independently of the year in which they were collected, allowing for a ready comparison of the typical number concentrations. Here, we can see that baseline fungal concentrations of 5–10s $L^{-1}$ are typical, with occasional periods of enhancement up to 100 $L^{-1}$ also being observed. Bacteria-like concentrations are typically of the order of a few per liter throughout.

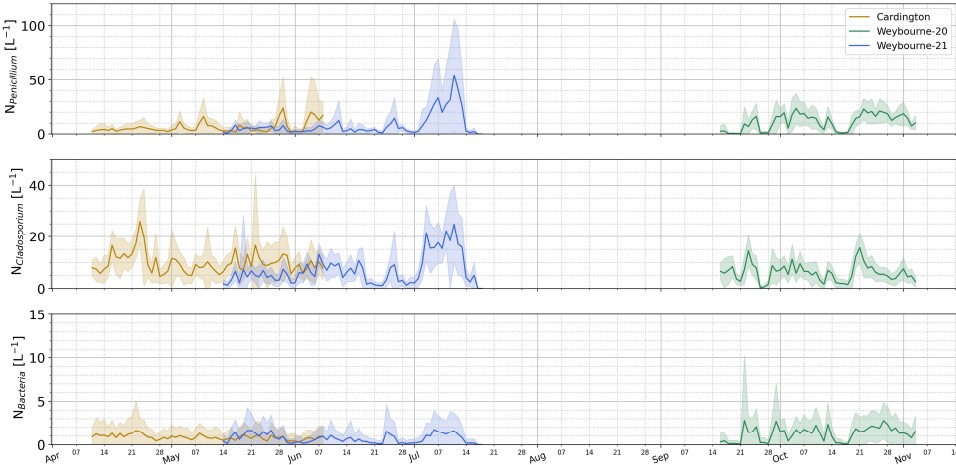

**Figure 9.** Comparison of class concentration time series at each site. The line represents the daily mean value; the shaded area is the standard deviation. All data have been offset to the same year for direct inter-comparison.

To further investigate the diurnal behavior of the different classes at each site, we first look at standard hourly diurnal averaging (Figure 10, top row), and then we further apply daily Z-score normalization over a daily rolling window to facilitate the direct comparison of the trends of each class, while minimizing the impact of episodic events and skew from day-to-day variations in concentrations (Figure 11, bottom row). Here, it can be seen that the *Penicillium*-like class displays a similar trend, featuring strong early-morning maxima and mid-day minima. This suggests that the *Penicillium*-like fungal emissions may be driven by changes in relative humidity, as observed in other studies [44–47].

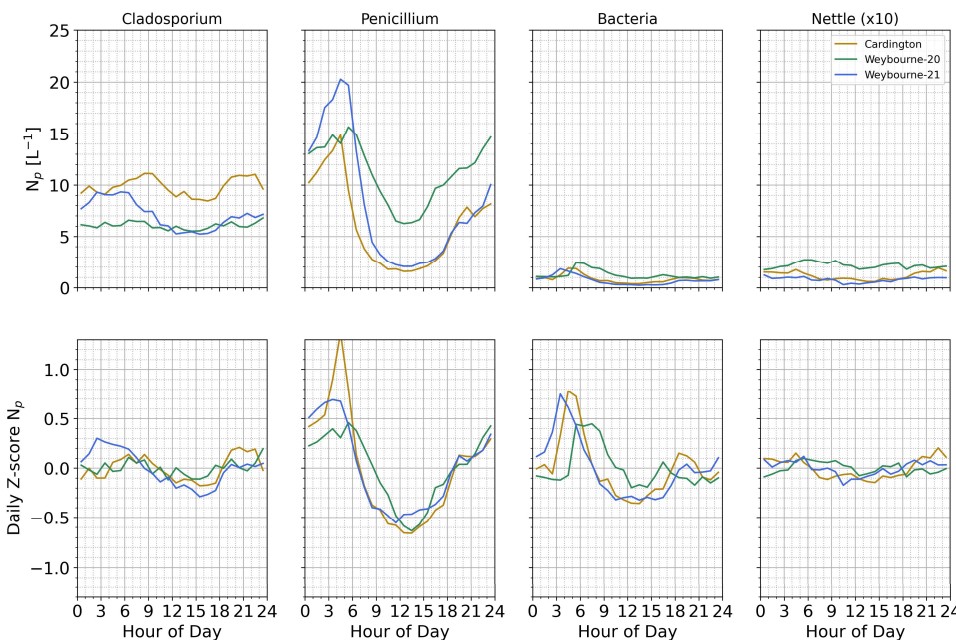

**Figure 10.** Hourly averaged diurnal variation in each class (columns) at each site. Top row displays the hourly average number concentrations; bottom row is processed over a daily window, where each day of observations is independently z-score normalized to minimize the impact of differing inter-day concentration variability while preserving the underlying diurnal trend. Note: nettle number concentration values (top row) have been increased by a factor of 10 for clarity.

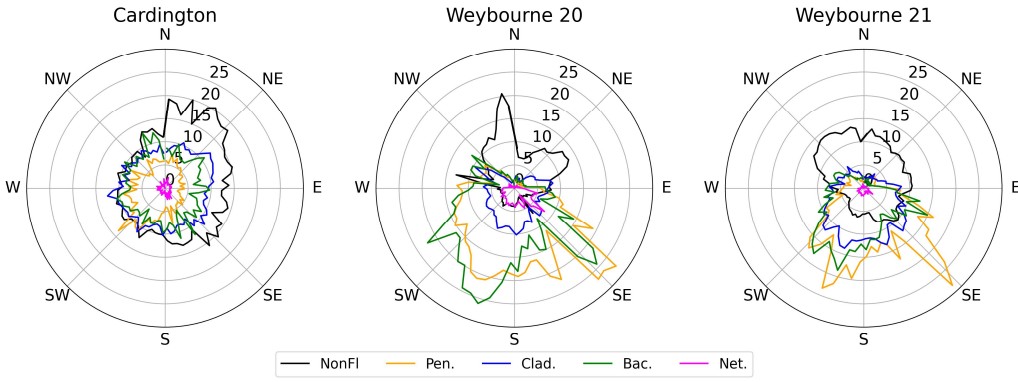

**Figure 11.** Mean class concentrations at each site as a function of wind direction. Concentric rings are spaced at 5 L$^{-1}$ intervals. Data are averaged over 5° bins. Note that all nettle and bacteria concentrations have been increased by a factor of 10 for clarity; non-fluorescent concentrations have arbitrarily been decreased by a factor of 100, 500, and 300 at each site, respectively, for comparison to the BioPM classes.

The *Cladosporium*-like class does not display a strong diurnal trend, whereas the trends are more apparent in the daily Z-score diurnal analysis. Here, it can be seen that *Cladosporium*-like particles peak at different times compared to the *Penicillium*-like particles, occurring earlier and later at the Weybourne 2021 and Cardington deployments, respectively; no observable diurnal trend was seen in the autumnal Weybourne 2020 deployment, possibly due to the reduced autumnal boundary layer height. While not strongly evident here, *Cladosporium* is generally expected to be most abundant in the afternoon, maximizing atmospheric transit in a turbulent boundary layer [38,39]; however, a slight preference for early evening was observed at the Cardington site. We advise caution in overinterpreting this result; the variations are small in all cases. Bacteria-like aerosols

display a slight preference for early morning emission, and nettle-like concentrations are reasonably constant throughout the day.

We now investigate the observed meteorological relationships for each class at each site. First, we investigate the influence of wind direction (Figure 11, note scaling factors in figure caption): at Cardington, there was no strong wind direction dependence observed in the *Cladosporium*- and bacteria-like classes, suggesting that there was no strong local source region; *Penicillium*-like aerosols displayed a preference from the south-west and west wind sectors, suggesting a source in that region. Contrary to the BioPM observations, non-fluorescent aerosols were more strongly correlated with winds from the north to south-eastern sectors. The areas immediately surrounding the site are rural and peri-urban in nature, with the towns of Bedford and Kempston to the north-west and west, respectively. Bedford, which is the larger of the two towns, does not appear to correlate with any of the BioPM observations, but there may be a slight correlation between Kempston and the *Penicillium*-like class.

For both of the Weybourne deployments, the picture is clearer; non-fluorescent aerosols tend to be most strongly observed from northern wind sectors, correlating with coastal/marine sectors and their associated emissions; *Penicillium*-like aerosols display a strong response from the SE, correlating to winds blowing along the coast to the site; all BioPM classes also display a strong response to winds from the SSW sector, correlating with rural/farmland and wooded areas in this direction.

Finally, we investigate the influence of meteorological parameters. Figure 12 shows the relationship between relative humidity (RH), temperature, and wind speed on the mean BioPM class concentrations. *Penicillium*-like aerosol displays the clearest and strongest variation to RH (top row), with concentrations increasing with increasing RH (approximate Pearson and Spearman correlation coefficients > +0.8); the other classes tend to display a relatively flat RH response, except for bacteria-like aerosols which demonstrate a positive relationship (approximate Pearson and Spearman correlation coefficients > +0.55).

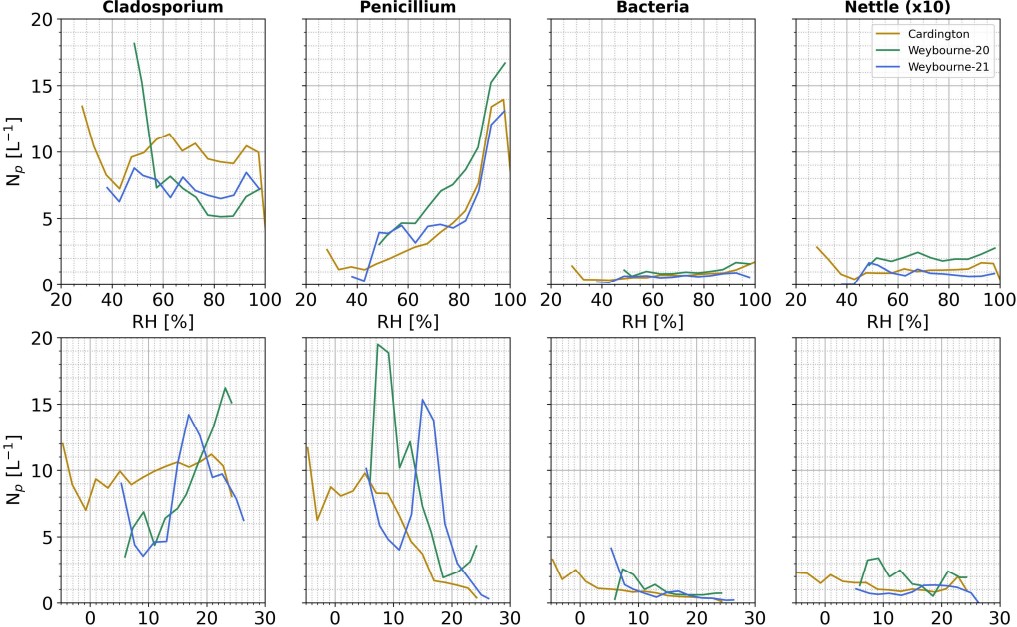

**Figure 12.** *Cont.*

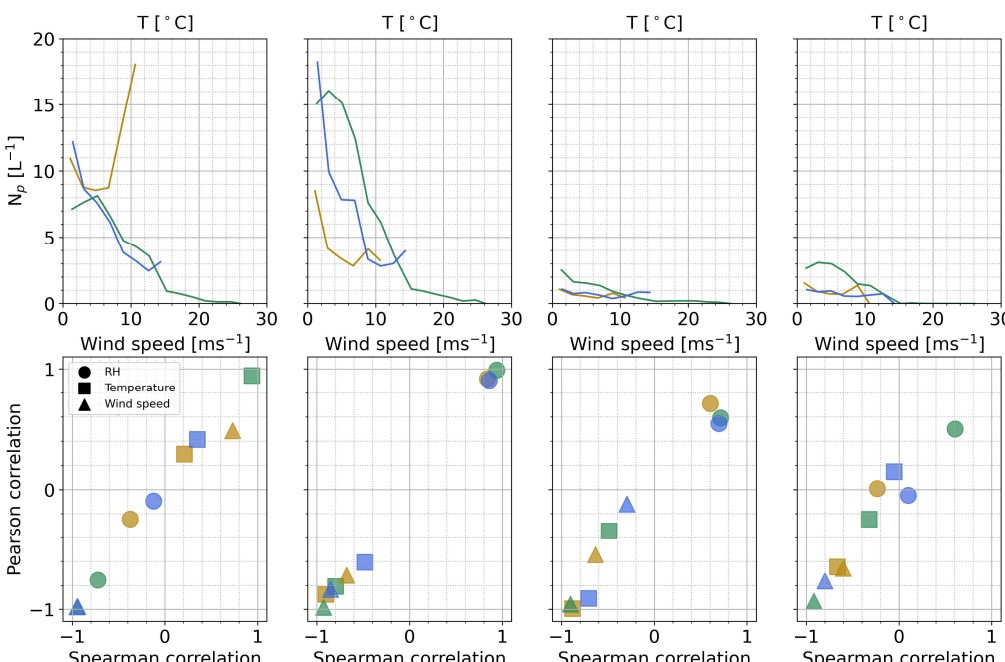

**Figure 12.** Influence of meteorological parameters on each class (columns) at each site. Top row: relative humidity; middle–top row: temperature; middle–bottom row: wind speed. Mean value shown. Bottom row: Pearson vs. Spearman correlation coefficients for each meteorological parameter (shape, see legend) and site; colors denote the site (see legend in nettle/RH panel). Note: nettle number concentration values have been increased by a factor of 10 for clarity.

The trends with temperature are less striking, featuring modal relationships in some cases, which likely correlates with the temperatures coinciding with the peak diurnal RH at each deployment. *Cladosporium*-like aerosols display a moderate to strong positive correlation, favoring warmer conditions (approximate Pearson and Spearman correlation coefficients +0.3–0.9); *Penicillium*- and bacterial-like aerosol both display moderate to strong negative relationships to temperature (approximate Pearson and Spearman correlation coefficients $<-0.5$ and $<-0.35$, respectively). The highest concentrations are typically observed at lower wind speeds, suggesting that the sources are local rather than arriving via long-range transport. *Cladosporium*-like aerosols display the opposite trend during the Cardington deployment; however, the data points corresponding to the highest observed wind speeds are limited to a single event around 27 April in this case. General moderate to strong negative correlation coefficients are observed in most cases.

## 4. Summary and Conclusions

In this study, we investigated the emissions of key BioPM species at peri-urban/rural and coastal sites in the UK during key seasons, exploiting the high time-resolution data afforded by real-time UV-LIF bioaerosol spectrometers to probe relationships with finer granularity than could be achieved with traditional detection methods. Here, we have demonstrated the utility of such an approach, in conjunction with a new dimensional reduction-based classification technique, to appraise emissions and investigate potential mechanisms and drivers.

We provide representative characteristic concentrations and behaviors of the trained BioPM classes at each site, where generally, the UMAP classifier detects similar particles at each site and deployment. This allows us to suggest baseline and typical diurnal behaviors for each class for further use by the wider community to estimate emission and exposure impacts in similar outdoor locations. The clear and expected RH-driven diurnal behavior of *Penicillium*-like fungal aerosols is a key result, supporting the method's utility. It is envisaged that finer granularity of classification would be possible with more sophisticated

UV-LIF spectrometers (e.g., Plair Rapid-E+ and Swisens Poleno), which feature greater spectral resolution, higher speed optoelectronics, and additional advanced capability such as fluorescence lifetime to expand on the identification of fluorophore composition; the referenced spectrometers also boast enhanced upper size detections of ~100 μm and inlet configurations more suited for pollen detection, facilitating investigations into pollen emissions. As such, the work in this paper using last-generation spectrometers represents a minimum capability in the rapidly growing field of real-time bioaerosol detection.

A clear result arising from this work is the quantification of the effects of rainfall on the enhancement of nighttime diurnal *Penicillium*-like aerosols, where we demonstrated that the emissions maxima of this fungal class may be significantly enhanced above baseline values under wet conditions. Rainfall acts only to enhance the quantity emitted; the timing of the emission appears to be consistently constrained independently of rainfall, driven by diurnal variation in relative humidity. Crucially, this result may not have been possible to observe with the low time-resolution common to traditional sampling methods. This enhancement may prove significant for nighttime air quality and exposure and warrants further investigation. Furthermore, this may impact allergies and respiratory conditions, especially if airmasses containing rainfall-enhanced fungal concentrations are mixed into indoor micro-environments via windows opened during warmer periods. Further long-term real-time BioPM monitoring is needed to investigate its emission and impacts on allergies and health outcomes; we suggest that such high time-resolution monitoring data could be of great value to understanding respiratory and allergenic syndromic surveillance health data, potentially leading to the development of improved intervention strategies and also informing policy surrounding net zero greenspace development.

Rainfall-enhanced fungal emissions such as those observed here may also impact climate via cloud–aerosol interactions. As discussed earlier, emissions of ice-active BioPM such as *Fusarium* have been demonstrated to be enhanced by rainfall [16]. While the classifier used in this study was not trained on *Fusarium*, there is no apparent reason why this should not be possible with appropriate training data to investigate *Fusarium* emissions. It has previously been shown that even a modest primary ice concentration of 0.01 L$^{-1}$ may lead to rapid supercooled cloud glaciation via secondary ice processes, leading to enhanced precipitation, where BioPM may contribute to the source of primary ice nucleating particles necessary to initiate this process [20]; this builds the narrative of the bioprecipitation hypothesis [23,24], where it has been speculated that rainfall-induced enhancement of ice-active BioPM yields an environment beneficial for the growth of plants and microorganisms via enhanced precipitation caused by bio-IN driven cloud glaciation, creating feedback where enhancement in rainfall creates an enhancement in bio-IN emissions. Widescale high time-resolution BioPM monitoring will be necessary to investigate this hypothesis via *big data* approaches due to the likely meso- to synoptic scale nature of the feedback, again showcasing the impacts and utility of the methods employed in this study to tackle big questions pertaining to BioPM impacts.

**Author Contributions:** Conceptualization, I.C., M.G., K.B. and D.T.; methodology, I.C.; software, I.C.; formal analysis, I.C.; investigation, I.C., D.T. and M.G.; resources, M.G.; experiments at ChAMBRe, D.M., S.D.P. and V.V.; data curation, I.C., K.B., S.D.P., V.V. and D.M.; writing—original draft preparation, I.C., M.G. and D.T.; funding acquisition, M.G., D.T. and I.C. All authors have read and agreed to the published version of the manuscript.

**Funding:** This research was funded by the NERC BIOARC programme, grant number NE/S002049/1. The ChAMBRe laboratory studies are part of a project that is supported by the European Commission under the Horizon 2020—Research and Innovation Framework Programme, H2020-INFRAIA-2020-1, project: ATMO-ACCESS Grant Agreement number: 101008004.

**Institutional Review Board Statement:** Not applicable.

**Informed Consent Statement:** Not applicable.

**Data Availability Statement:** The data presented in this study are available on request from the corresponding author.

**Acknowledgments:** We acknowledge Paolo Prati and Mirca Zotti, University of Genova, for providing access to ChAMBRe and fungal strains, and Marco Brunoldi and Elena Gatta, University of Genova, for the technical support during the experiments at ChAMBRe. The authors would like to acknowledge the Atmospheric Measurement and Observation Facility (AMOF), a Natural Environment Research Council (UKRI-NERC) funded facility, for providing access to the Weybourne Atmospheric Observatory and site meteorological data. We acknowledge Jeremy Price and James McGreggor for providing access to the Meteorological Research Unit, Cardington.

**Conflicts of Interest:** The authors declare no conflict of interest.

## Abbreviations

The following abbreviations are used in this manuscript:

| | |
|---|---|
| BioPM | Biological particulate matter |
| ChAMBRe | Chamber for Aerosol Modeling and Bio-aerosol Research |
| CMOS | Complementary metal-oxide semiconductor |
| FT | Forced trigger |
| LPM | Litres per minute |
| MBS | Multiparameter bioaerosol spectrometer |
| MIDAS | Met Office Integrated Data Archive System |
| UMAP | Uniform manifold approximation and projection for dimension reduction |
| UV-LIF | Ultraviolet light-induced fluorescence |
| WIBS | Wideband integrated bioaerosol spectrometer |

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
