# Peer review of "Towards a UK Airborne Bioaerosol Climatology: Real-Time Monitoring Strategies for High Time Resolution Bioaerosol Classification and Quantification"

_atmosphere, doi:10.3390/atmos14081214_

Round 1

Reviewer 1 Report

General comments

This paper introduces the UV-fluorescent detection device which can monitor several kinds of atmospheric bioaerosols classifiably and analyze the variations of bioaerosols during the three survey periods. The authors found the dual changes of airborne fungi and bacteria and the relations of microbial abundances to environmental factors. The monitoring observation targeting bioaerosols are expected strongly to be essential technology for elucidating the pathogenic impacts on human health via bioaerosols dispersion as well as the atmospheric microbial roles in climate changes. Although, the results and main discussion would be impressive and meaningful, I think there are some issues as format level in this paper. Accordingly, these issue may disturb some readers understand this paper values.

        I have to recommend this manuscript need some revisions before accept in this journal .

Some major comments:

1.     The Results and Discussion section contains some overlapped topics and redundancies. For example, the biological-particle variations during the three survey periods are discussed separately, but some discussions are overlapped. This study aims are thought to be 1) establishing the measurement techniques using MBS with  the original classification algorithms, 2) measuring the biological aerosols by the established MBS systems, and 3) discussing the relations of the biological aerosol abundances to the environmental factors. The Result and Discussion section has to be shortened to focus on the upper topics.

2.     With relating the comment 1, this paper introduce the biological classification mechanisms are introduced in the Material section, but this topic is thought to contain the originality contents for this paper. Accordingly, this parts can be moved to the Result and Discussion section and shorten to be clearer description as the result and discussion topics.

3.     Statistical analyses are needed for discussing the relations of the biological aerosol abundances to the environmental factors. In particular, correlation values have to be calculated and evaluated using  statistical analyses.

4.     I think this paper used a lot of abbreviations, which make hard understand this paper. Could the authors change some abbreviations more meaningful one or avoid the abbreviation description?

Some minor comments:

L120: Please insert one paragraph explaining the aim of this study.

L130: Each wave length used in this study have to be explained in detail.

L123: How is “MB spectrometer”?

L133: What is LPM?

L139: What is CMOS?

L187-L225: This part contain the introduction contents, which should be moved to the Introduction section and have to be shorten as Materials and Methods contents.

L226-L335: These sections have to be shorten to move to the Result and Discussion section. All readers of the journal Atmosphere are not computer scientist, so I recommend this parts changed to the more understandable for general atmospheric researchers.

L337-L338: This explanation can be removed.

L339-L433: These parts should be combined to indicate the sequential changes during the three survey period regardless of sites as Fig. 10. Additionally, the seasonal variations of fungi and bacteria have to be discussed with some reasons referring some papers (ex. Maki et al., Atmospheric Environment, 16, 2023. doi.org/10.1016/j.atmosenv.2023.119726.)

L434- L510: As described, the statistical analyses are needed for the discussion of relations between biological aerosols and environmental factors.

Reviewer 2 Report

1.          This study used newly UV-LIF spectrometer to monitoring bioaerosol in key season. The concept is innovated and original. The effort on overcoming difficulties of conventional sampling techniques is appreciated. However, there is still some scientific problems should be clarified.

2.          The present manuscript did not show the design of aerosol collector. The sampling flow was set on 0.2 LPM (as Line 133), is it efficient to collect bioaerosol from atmospheric environment? How about the facing velocity of the collector? The sampling result of aerosol collector which apply low sampling flow may be interfered by cross wind. Please provide more information.

3.          The appearance difference (shape, diameter, color, edge, etc.) of Penicillium and Cladosporium could be identified by microscopic investigation with trained technician. What kind of principles and methods used to train your 2D UMAP model? How about the deviation and precision based on microscopic comparison? In addition, there are still much biological origin particle suspended in the atmospheric environment (Aspergillus spp., pollen, insects, etc.) How this technology made the creditable classification? Please provide more information.

4.          Considering of sampling efficiency, challenging experiments of real viable bioaerosol in environmental-controlled chamber is helpful. Is there any more preliminary aerodynamic data (such as aerodynamic diameter, the intensity of fluorescence, etc.) to prove the performance of this aerosol collector/spectrometer in ChAMBRe simulation chamber?

5.          Regarding to health risk of respiratory system, the viability of the bioaerosols is important issue. Is there any contribution of this UV-LIF spectrometer can be attributed in this study?

Round 2

Reviewer 1 Report

This paper has been revised as an article paper in dependence on the reviewer's comments.

I would like to recommend this paper is published in this journal.